# Understanding Boolean Function Learnability on Deep Neural Networks: PAC Learning Meets Neurosymbolic Models

**Marcio Nicolau**                                        MARCIO.NICOLAU@INF.UFRGS.BR
**Anderson R. Tavares**                                        ARTAVARES@INF.UFRGS.BR
**Pedro Henrique da Costa Avelar**                                        PHCAVELAR@INF.UFRGS.BR
**João M. Flach**                                        JMFLACH@INF.UFRGS.BR
*Institute of Informatics, Federal University of Rio Grande do Sul, Brazil*
**Zhiwei Zhang**                                        ZZ59@RICE.EDU
*Department of Computer Science, Rice University, Houston, USA*
**Luis C. Lamb**                                        LAMB@INF.UFRGS.BR
*Institute of Informatics, Federal University of Rio Grande do Sul, Brazil*
**Moshe Y. Vardi**                                        VARDI@CS.RICE.EDU
*Department of Computer Science, Rice University, Houston, USA*

**Editors:** Leilani H. Gilpin, Eleonora Giunchiglia, Pascal Hitzler, and Emile van Krieken

## Abstract

Computational learning theory states that many classes of boolean formulas are learnable in polynomial time. This paper addresses the understudied subject of how, in practice, such formulas can be learned by deep neural networks. Specifically, we analyze boolean formulas associated with model-sampling benchmarks, combinatorial optimization problems, and random 3-CNFs with varying degrees of constrainedness. Our experiments indicate that: (i) neural learning generalizes better than pure rule-based systems and pure symbolic approach; (ii) relatively small and shallow neural networks are very good approximators of formulas associated with combinatorial optimization problems; (iii) smaller formulas seem harder to learn, possibly due to the fewer positive (satisfying) examples available; and (iv) interestingly, underconstrained 3-CNF formulas are more challenging to learn than overconstrained ones. Such findings pave the way for a better understanding, construction, and use of neurosymbolic AI methods.

## 1. Introduction

The growing need for Artificial Intelligence (AI) systems that integrate reasoning and learning has been pointed out by Turing Award winner and machine learning pioneer Leslie Valiant as a key challenge for computer science (Valiant, 2013, 2021). Several deep learning (DL) methods and technologies have impacted AI research and defined new large-scale technologies in various domains (Kahneman et al., 2020; LeCun et al., 2015; Schmidhuber, 2015). More recently, the need for methods that combine machine learning and symbolic reasoning has been identified as a relevant challenge in industry and academia (Kahneman et al., 2020; Raedt et al., 2020; Raghavan, 2019; Silver et al., 2022; Sen et al., 2022; Mao et al., 2021).

However, there remain several challenges to bridge the gap between theoretical and practical advances in machine learning that would allow such effective integration (d'Avila Garcez and Lamb, 2023; d'Avila Garcez et al., 2019; Mao et al., 2019; Marcus, 2020; Raedt et al., 2020; Raghavan, 2019). In AI conferences and academic debates, including

the Montreal AI debates of 2019, 2020, and 2022 and the AAAI-2021 conference, leading researchers including DL pioneers Yoshua Bengio, Geoffrey Hinton and Yann LeCun have singled out the effective development of integrated reasoning mechanisms to DL systems as essential to machine learning progress (Kahneman et al., 2020). In this setting, our work contributes to establishing foundations for the development of neurosymbolic methods by showing to which extent boolean learnability (mainly studied under a theoretical perspective) is practically feasible in a DL framework. Moreover, to integrate reasoning and learning, one has to show that DL can effectively learn symbolic (and logical) concepts, which we show in this paper. This interplay of neural learning and boolean concepts thus contributes to analyse a foundational component of logical methods in computer science and machine learning, where boolean functions abound in many domains (Knuth, 2008; Valiant, 2013).

To achieve such integration, one has to consider questions still open in deep learning, such as effective algorithms for reasoning and learning over classes of boolean formulas, learnable in polynomial time according to computational learning theory (Valiant, 1984). These outstanding questions to be addressed particularly refer to effective experimentation on classes of boolean functions (Kearns et al., 1994b). Further, developing efficient learning algorithms for boolean formulas remains challenging in AI (Valiant, 2013).

Seeking to respond to the above challenges, this paper offers key contributions: (i) our experiments on model sampling benchmarks show that neural learning generalizes better than pure rule-based systems and pure symbolic approach; (ii) experiments on combinatorial optimization problems contribute to the integration of learning and reasoning since we show that relatively small and shallow neural networks can learn several families of boolean functions that encode such problems; (iii) smaller formulas seem harder to learn, possibly due to the fewer positive (satisfying) examples available; and (iv) interestingly, underconstrained 3-CNF (formulas in conjunctive normal form with 3 literals per clause) are more challenging to learn than overconstrained ones. We also show that simple neural network models can reproduce encodings of combinatorial optimization problems. Source code and relevant datasets are publicly available[1].

## 2. On Deep Boolean Function Learnability

This section presents the fundamental concepts and definitions on boolean learnability, as well as related work. In the seminal PAC learning paper, Valiant (1984) highlighted the importance of knowledge representation and the relationship with logic and the design of machine learning systems: "...[the] remaining design choice that has to be made is that of knowledge representation. Since our declared aim is to represent general knowledge, it seems almost unavoidable that we use some kind of logic rather than, for example, formal grammars or geometrical constructs. [...] we shall represent concepts as Boolean functions of a set of propositional variables. The recognition algorithms that we attempt to deduce will be therefore Boolean circuits or expressions." Over the years, machine learning, and knowledge representation and reasoning too followed separate methodologies (d'Avila Garcez et al., 2009; Besold et al., 2022).

---

1. https://github.com/machine-reasoning-ufrgs/mlbf and https://github.com/zzwonder/Valiants-Algorithm-for-Learning-CNF-formulas.

We use Classical Propositional Logic (*PL*) with the usual set of logical connectives $\{\neg, \wedge, \vee\}$. A boolean formula (*BF*) is a *string* in the *PL* language. It can be recursively defined as: (i) a literal, which is a variable or a negated variable, (ii) a conjunction, which is a set of *BF*s separated by $\wedge$, or (iii) a disjunction, which is a set of formulas separated by $\vee$. A boolean function is the function $f : \{0, 1\}^k \rightarrow \{0, 1\}$ defined by a boolean formula. It takes as arguments an assignment for every variable in the formula and returns a value - zero or one - according to the semantics of the connectives in the formula. An evaluation of a *BF* is the result of computing the boolean function associated with it. A Conjunctive Normal Form (CNF) is a specific type of *BF*, where the formula is expressed as a conjunction of one or more clauses, which are disjunctions of literals. A k-CNF is a CNF where each clause has no more than k literals. Every *BF* can be transformed into a logically equivalent CNF; and every CNF can be converted into a logically equivalent k-CNF (for $k \geq 3$).

Given a Boolean Formula, the Boolean Satisfiability problem (SAT) is defined as finding an assignment of the variables where the *BF* evaluates to true, or to provide the proof that no satisfying assignment exists. Usually, SAT solvers take as input a *BF* in the CNF format. The SAT problem is foundational in computer science and numerous practical problems.

**Related Work**   Computational learning theory presents hardness results on the learnability of boolean functions related to certain classes of problems, such as cryptography (Rivest, 1991), robust learning (Gourdeau et al., 2019) and distribution learning (Kearns et al., 1994a). It also presents positive results on polynomial-time learnability of boolean formulas, which are of our interest. In the probably approximately correct (PAC) framework, (Valiant, 1984) shows that conjunctive normal formulas with a bounded number of literals per clause (k-CNFs) are learnable in general, but not mentioning neural networks. Artificial Neural networks (ANNs) are universal learners of boolean formulas (Blum, 1989; Steinbach and Kohut, 2002), since classical perceptrons can be arranged to implement any logical gate and such gates can be arranged to implement any boolean formula, also with the possibility of extracting boolean formulas from trained neural networks (Tsukimoto, 1997). Moreover, even single-hidden-layer networks are universal boolean function learners (Anthony, 2010), although the worst-case number of neurons in the hidden layer is exponential on the number of inputs. Anthony (2010) further provides lower bounds on the sample complexity of boolean functions, relating the VC-dimension (Vapnik and Chervonenkis, 1971), with the tolerance (expected error margin) and confidence of the PAC learning framework. The width of the ANN can be traded off by depth to alleviate the worst-case requirement for the number of neurons (Anthony, 2010). Other ANNs have also been proved to universally implement boolean formulas, such as the binary pi-sigma network (Shin and Ghosh, 1991), the binary product-unit network (Zhang et al., 2011)

A body of empirical work followed the (positive) theoretical results on the learnability of boolean functions: Miller (1999) shows that parity and multiplier functions are efficiently learnable, Franco and Anthony (2004); Franco (2006); Franco and Anthony (2006) analyse complexity metrics related to the generalisation abilities of boolean functions implemented via neural networks, Subirats et al. (2006, 2008) and Zhang et al. (2003) propose algorithms for learning boolean circuits with thresholding neural networks, while Prasad and Beg (2009) study pre-processing techniques for using ANNs to learn boolean circuits, while Beg et al. (2008) studies approximating boolean functions' complexity using ANNs. Pan and Srikumar

(2016) showcases how ANNs with ReLU activation implement boolean functions much more compactly than with threshold linear units, and Daniely (2017) analyses how stochastic gradient descent learns function classes that comprise a relevant fraction of functions that are polytime PAC learnable. However, to the best of our knowledge, this is the first study on the learnability using neural networks on boolean formulas encoding combinatorial optimization problems and on the relation of learnability and constrainedness of random 3-CNFs.

## 3. Learnability Experiments

This section presents learnability experiments in various scenarios. First we analyze the learning capabilities of MLPs in combinatorial optimization problems (Sec. 3.1). Following, we investigate the difficulty of MLPs learning on synthetic 3-CNF formulas and compare them with previous results on satisfiability (Sec. 3.2). Finally, we compare symbolic and neural learning (Sec. 3.3). We have also compared multi-layer perceptrons (MLPs) and decision trees (DTs) on model sampling benchmarks in Appendix B.

In our experimental setup, we use scikit-learn (Pedregosa et al., 2011) implementation of fully-connected MLPs as our deep NN. We use Adam optimization (Kingma and Ba, 2014) with recommended parameters (learning rate $= 10^{-3}$, $\beta_1 = 0.9$, $\beta_2 = 0.999$), 200 training epochs and L2 regularisation term $= 10^{-4}$. Experiments in Appendix. B and Sec. 3.1 are performed with ReLU activation and a neural network with two hidden layers containing 200 and 100 neurons, respectively. Sec. 3.2 shows experiments with variations on activation functions and number of neurons in a single hidden layer. Sec. 3.3 shows experiments with the deep NNs as in Appendix B and Sec. 3.1; for the symbolic counterpart, a special case of conjunction, 3-CNF PAC-learnable algorithm (Valiant, 1984; Kearns and Vazirani, 1994). Experiments in this section were performed on a computer with a 6-core (12 threads) Intel Core i7-8700 CPU @ 3.20GHz and 32 GiB DDR4 RAM.

### 3.1. Combinatorial Optimization Problems

Preliminary experiments (Appendix B) showed that multi-layer perceptrons (MLPs) had perfect 5-fold cross-validation accuracy, outperforming decision trees (DTs) on model sampling benchmarks, which encode interesting practical problems. This section further analyses the learning capabilities of MLPs on other classes of boolean formulas: those encoding the decision version of NP-complete combinatorial optimization problems.

We choose versions of graph colouring (is there a way to colour the graph vertices with k colours such that adjacent vertices have different colours?) and clique (is there a complete subgraph with k vertices?) (Arora and Barak, 2009) as our problems. We refer to these as k-GCP and k-clique hereafter. The number of satisfying assignments of formulas encoding these problems is the number of k-colourings and k-cliques, respectively. GCP instances are on flat and morphed graphs, retrieved from SATLIB[2]. GCP flat instances are 3-colourable quasi-random graphs. Different graphs are generated with the same number of nodes and edges, whose connectivity is arranged to make them difficult to solve by the Brelaz heuristic (Hogg, 1996). GCP morphed instances are 5-colourable graphs, constructed by merging

---

2. https://www.cs.ubc.ca/~hoos/SATLIB/benchm.html. Follow the "description" link on the page for an explanation on flat vs morphed graphs.

regular ring lattices, whose vertices are ordered cyclically and each vertex is connected to its 5 closest in this ordering, with random graphs from the (Erdös and Rényi, 1959) model. An $r$-morph of two graphs $G_1 = (V, E_1)$ and $G_2 = (V, E_2)$ is a graph $G = (V, E)$ where $E$ contains all edges in $E_1 \cap E_2$, a fraction $r$ of edges in $E_1 - E_2$, and a fraction $1 - r$ of edges in $E_2 - E_1$. These GCP instances are satisfiable by construction.

Our k-clique instances are generated on random $G(n, p)$ graphs (Erdös and Rényi, 1959) with CNFgen (Lauria et al., 2019), where we control the number of nodes $n$ and the probability of each edge $p$ to result in the desired number of k-cliques (Bollobás and Erdös, 1976). In particular, we generate graphs with $n = 50, 100$ and 150 nodes, aiming for 500 3-cliques on average. This gives $p = 0.2944, 0.1457$ and 0.0968, respectively. All these formulas are also satisfiable by construction. As in Appendix B, for each formula, we aim to generate a dataset with 500 positive and 500 negative samples. Our deep NN is an MLP with two hidden layers containing 200 and 100 neurons, respectively. For each problem size (i.e. #nodes and #edges), $|S|$ denotes the number of datasets generated, i.e., for how many formulas we were able to generate samples. We also show the size of the resulting formulas (#variables and #clauses), the mean and minimum accuracies of the MLP on the 5-fold CV across all formulas and the ratio of formulas with perfect (i.e. 100%) accuracy.

Table 1: Learnability results on combinatorial optimization problems. $c/v$ is the clause-to-variable ratio of the resulting formulas. $|S|$ denotes the number of formulas for which Unigen2 was able to generate samples. % perfect is percent of formulas for which the MLP achieved 100% accuracy. 3-fGCP stands for 3-GCP on flat graphs, 5-mGCP stands for 5-GCP on morphed graphs with different morph ratios ($r$). 3-clique problems are on $G(n, p)$. On 3-clique formulas, * denotes values on average because actual values vary for each individual formula.

| Problem | #nodes | #edges | $|S|$ | #vars | #clauses | c/v | Mean acc (%) | Min acc (%) | % Perfect |
|---|---|---|---|---|---|---|---|---|---|
| 3-fGCP | 30 | 60 | 100 | 90 | 300 | 3.33 | 99.92 | 99.46 | 65.00 |
| 3-fGCP | 50 | 15 | 998 | 150 | 545 | 3.63 | 99.97 | 99.17 | 89.00 |
| 3-fGCP | 75 | 80 | 100 | 225 | 840 | 3.73 | 99.98 | 99.40 | 94.00 |
| 3-fGCP | 100 | 239 | 100 | 300 | 1117 | 3.72 | 99.97 | 99.21 | 88.00 |
| 3-fGCP | 125 | 301 | 100 | 375 | 1403 | 3.74 | 99.98 | 99.21 | 90.00 |
| 3-fGCP | 150 | 360 | 99 | 450 | 1680 | 3.73 | 99.98 | 99.60 | 91.90 |
| 3-fGCP | 175 | 417 | 82 | 525 | 1951 | 3.72 | 99.84 | 98.44 | 73.20 |
| 3-fGCP | 200 | 479 | 64 | 600 | 2237 | 3.73 | 99.82 | 98.24 | 64.10 |
| 5-mGCP, r=1 | 100 | 400 | 94 | 500 | 3100 | 6.20 | 100.00 | 100.00 | 100.00 |
| 5-mGCP, r=0.5 | 100 | 400 | 100 | 500 | 3100 | 6.20 | 100.00 | 100.00 | 100.00 |
| 5-mGCP, r=0.25 | 100 | 400 | 100 | 500 | 3100 | 6.20 | 100.00 | 100.00 | 100.00 |
| 5-mGCP, r=0.125 | 100 | 400 | 100 | 500 | 3100 | 6.20 | 100.00 | 100.00 | 100.00 |
| 5-mGCP, r=$2^{-4}$ | 100 | 400 | 100 | 500 | 3100 | 6.20 | 99.95 | 99.47 | 75.00 |
| 5-mGCP, r=$2^{-5}$ | 100 | 400 | 100 | 500 | 3100 | 6.20 | 99.79 | 99.20 | 26.00 |
| 5-mGCP, r=$2^{-6}$ | 100 | 400 | 100 | 500 | 3100 | 6.20 | 99.93 | 99.47 | 66.00 |
| 5-mGCP, r=$2^{-7}$ | 100 | 400 | 100 | 500 | 3100 | 6.20 | 100.00 | 99.84 | 98.00 |
| 5-mGCP, r=$2^{-8}$ | 100 | 400 | 100 | 500 | 3100 | 6.20 | 99.99 | 99.84 | 96.00 |
| 5-mGCP, r=0 | 100 | 400 | 1 | 500 | 3100 | 6.20 | 100.00 | 100.00 | 100.00 |
| 3-clique | 50 | 360.64* | 100 | 150 | 10091.16* | 67.27 | 100.00 | 99.88 | 97.00 |
| 3-clique | 100 | 721.22* | 100 | 300 | 42695.94* | 142.32 | 100.00 | 99.76 | 98.00 |
| 3-clique | 150 | 1081.74* | 100 | 450 | 97790.55* | 217.31 | 100.00 | 99.88 | 96.00 |

Table 1 shows the results (there is a single instance of 5-GCP morphed formula with $r = 0$ because this morph rate results in a graph using all edges from the regular ring lattice and no edges from the random graph). In general, the mean and minimum accuracies are very close to, but not 100% throughout formulas on all sizes and problems, which means that

the MLP is a good approximator of these formulas. The metric that varies most across sets of formulas is the ratio of perfectly learned formulas (% perfect). Thus we use this metric as a proxy for learnability. Formulas encoding 3-GCP on flat graphs were more challenging on the smallest and largest graphs (30 and 200 nodes, respectively), with % perfect rates around 65%, whereas it remained close to 90% on the other graph sizes. Regarding the number of variables, these are respectively the smallest and largest sets of formulas in this experiment. We further investigate the difficulty imposed by different number of variables in Section 3.2.

In 3-clique problems, the ratio of perfectly-learned formulas is very similar and close to 100% across all graph sizes. This suggests that formulas encoding problems over random $G(n,p)$ graphs are easier to learn. 5-GCPs on morphed graphs showed an interesting behaviour of the ratio of perfectly-learned formulas on different morph ratios $r$: it is 100% on the largest 4 ratios, falling up to 26% on intermediate ratios and rising again to 100% on $r = 0$. Higher $r$ yields graphs with more edges from a random graph, which are easier to learn, as per the 3-clique experiments. On the other hand, small $r$ yields graphs with more edges from the ring lattice, whose regular structure results in easier GCP-encoding formulas. Mixing the structure of random and regular graphs with intermediate morph ratios yields the most challenging GCP-encoding formulas.

## 3.2. Satisfiability vs. Learnability on 3-CNF Formulas

This section investigates the learning performance of MLPs according to the size of the boolean formula. In addition, we assess the learning capabilities of deep neural networks regarding the constrainedness (clause-to-variable ratio, or $c/v$ for short) of formulas. We aim to verify whether hard-to-satisfy formulas are hard to learn. A large body of work has investigated how to generate hard-to-satisfy boolean formulas (Cheeseman et al., 1991; Crawford and Auton, 1996; Selman et al., 1996), showing that random 3-CNF formulas have a phase transition region associated with $c/v$. Formulas are easy to prove satisfiable when $c/v$ is below the phase transition region (underconstrained) and are easy to prove unsatisfiable when $c/v$ is above the phase transition region (overconstrained). We denote the constrainedness of the sweet spot between the under- and overconstrained, as "on phase". Formulas "on phase" are the hardest to solve. Table 2 depicts the $c/v$ of phase transition regions according to the number of variables, from 10 to 100.

Table 2: Satisfiability phase transition on random 3-CNFs according to the number of variables. Source is the paper where $c/v$ was obtained: CA (Crawford and Auton, 1996) or SML (Selman et al., 1996).

| #variables | 10 | 20 | 30 | 40 | 50* | 60 | 70 | 80 | 90 | 100* |
|---|---|---|---|---|---|---|---|---|---|---|
| Phase $c/v$ | 5.500 | 4.550 | 4.433 | 4.375 | 4.360 | 4.317 | 4.300 | 4.287 | 4.289 | 4.310 |
| Source | CA | CA | CA | CA | SML | CA | CA | CA | CA | SML |

In this section we investigate how hardness in satisfiability relates to hardness in learnability on random 3-CNF formulas. Our formulas range from 10 to 100 variables in increments of 10. These are relatively small to what SOTA SAT solvers routinely solve. This is on

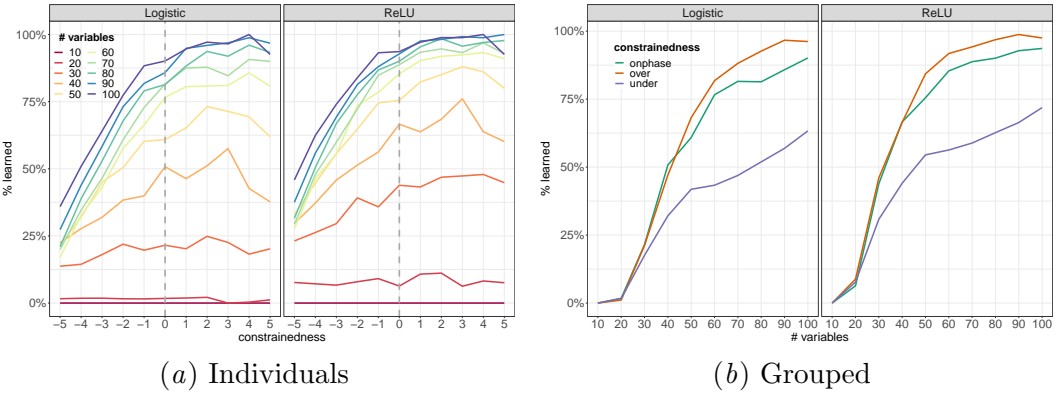

$(a)$ Individuals $\qquad\qquad$ $(b)$ Grouped

Figure 1: Percent of (perfectly) learned formulas, i.e. with 100% accuracy, according to the number of variables and constrainedness. In 1(a), the constrainedness appears in the x axis: underconstrained from -5 to -1, on phase at the vertical line on 0 and overconstrained from 1 to 5. In 1(b), the number of variables is in the x axis and each line shows the average % of learned formulas for constrainedness grouped into under, on phase and overconstrained.

purpose because very large formulas were easily learned (Appendix B) and the set with the smallest 3-GCP formulas was the hardest among sets with varying number of variables (Section 3.1).

For each number of variables, we generate sets of formulas with 11 values of constrainedness: 1 set on phase, 5 underconstrained sets (with on-phase $c/v$ subtracted by $0.1, 0.2, \ldots, 0.5$), and 5 overconstrained sets (with on-phase $c/v$ added by $0.1, 0.2, \ldots, 0.5$). Each set contains 1000 formulas. We use CNFgen (Lauria et al., 2019) to generate a total of $10 \times 11 \times 1000 = 110,000$ satisfiable boolean formulas in this study (satisfiability was verified with MiniSAT (Sorensson and Een, 2003)).

We assess the learnability of each generated formula as follows: train a single-hidden-layer neural network on the dataset generated for the formula, varying the number of hidden-layer neurons from 1 to 256 in powers of 2, and check how many neurons are sufficient to "learn" the formula perfectly (with 100% accuracy on the 5-fold cross validation). If perfect accuracy is not achieved with 256 neurons, we say that the formula has not been (perfectly) learned.

We test neural networks with ReLU and logistic (sigmoid) activation functions. Figure 1 shows, for each activation function, the percent of perfectly-learned formulas according to the number of variables and constrainedness, which we depict as ranging from -5 denoting the least constrained, 0 being on phase transition (hardest solubility) and +5 denoting the most constrained.

Interestingly, the smaller the formulas, the harder it is to learn them. In fact, no 10-variable formula was perfectly learned at all, regardless of the constrainedness (Fig. 1(a)). The ReLU neural network learned more formulas overall compared to logistic, specially with fewer variables. This is aligned with experimental (Nair and Hinton, 2010; Dahl et al., 2013)

and theoretical (Fiat et al., 2019) evidence favouring ReLU over other activation functions in deep learning.

Figure 1 also shows that as constrainedness increases up to a point, more formulas are learned. Beyond such point, learnability slightly drops for all formulas, except those with 40 variables, where the drop is sharp on both activations. The point of highest learnability never coincides with the point of hardest solubility (the phase transition point, highlighted by the vertical dashed line in Fig. 1(a)).

Figure 1(b) highlights the similarity on the behaviour of onphase and overconstrained formulas as the number of variables increase, in contrast with underconstrained formulas. Learnability of on phase and overconstrained formulas grows almost identically with less than 40 variables. Beyond this point, overconstrained formulas become easier to learn. Alongside the percentage of perfectly-learned formulas, we assess the network complexity by measuring the average number of neurons needed to learn the formulas. Figure 2 shows the average number of neurons for each number of variables and constrainedness.

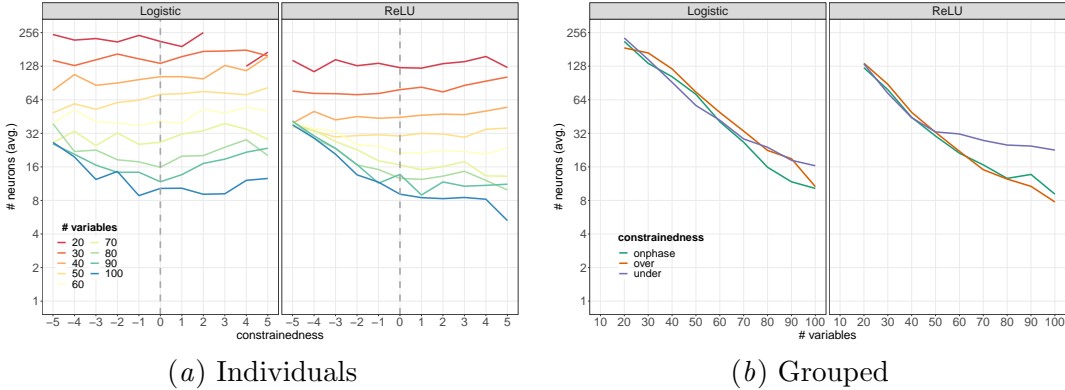

($a$) Individuals  ($b$) Grouped

Figure 2: Average number of neurons required to learn the formulas by number of variables and constrainedness. No line with 10 variables appears because no such formulas were perfectly learned at all (see Fig. 1). In 2($a$), the constrainedness appears in the x axis: underconstrained (-5 to -1), on phase (0) and overconstrained (1 to 5). The discontinuity with 20 variables and logistic activation happened because no formula with constrainedness=3 was learned. In 2($b$), the number of variables is in the x axis and each line shows the average number of neurons for constrainedness grouped into under, on phase and overconstrained.

Smaller formulas require more neurons to be learned. With logistic activation, the required number of neurons varies only slightly with few variables, regardless of the constrainedness. From 80 or more variables, the required number of neurons seem to relate with the difficulty in satisfiability: easy-to-solve formulas (either low or high constrainedness) require more neurons to learn, whereas hard-to-satisfy formulas (on phase transition) require less neurons. The stability of the number of neurons required to learn smaller formulas regardless of their constrainedness also occurs with ReLU. Moreover, the number of neurons required to learn formulas with 30 to 100 variables is very similar for the smallest constrainedness. Different

from logistic activation, where the average number of neurons is the highest for under- and overconstrained formulas, with ReLU activation, the overall trend is a monotonic decrease in the number of neurons as the constrainedness increases.

Figure 2(b) further shows that networks with logistic activation behave similarly when formulas are grouped by their constrainedness, i.e., requiring fewer neurons for larger formulas. On ReLU networks, however, the required number of neurons for underconstrained formulas decreases much slower with more than 50 variables compared to on phase and overconstrained. Fig. 2(b) also highlights that ReLU neural networks are more efficient in the sense of requiring less neurons to learn on-phase or overconstrained formulas. On larger underconstrained formulas, logistic neural networks are more efficient. We remark that these results apply to the formulas that have been (perfecly-)learned, as ReLU neural nets were able to effectively learn more formulas (Fig. 1), regardless of constrainedness.

### 3.3. Symbolic vs. Neural Network Learning

Table 3: Results of Valiant's algoritm (VA) (Valiant, 1984) vs the deep NN. Values present the average with std. deviation in parenthesis. #pos and #neg are the number of positive and negative assignments. *Count* is the average number of solutions of the set. $c/v$ is the clause-to-variable ratio (all formulas have 100 variables). $p/n$ is the positive-to-negative ratio of the assignments.

| Statistics / Clauses | 350 | 400 | 420 | 431 | 450 |
|---|---|---|---|---|---|
| VA Accuracy | 0.62 (0.04) | 0.81 (0.12) | 0.91 (0.10) | 0.91 (0.09) | 0.95 (0.07) |
| NN Accuracy | 0.99 (0.00) | 0.99 (0.00) | 1.00 | 1.00 (0.00) | 1.00 (0.00) |
| VA F1-Score | 0.37 (0.12) | 0.74 (0.20) | 0.88 (0.15) | 0.89 (0.13) | 0.92 (0.10) |
| NN F1-Score | 0.99 (0.00) | 0.99 (0.00) | 1.00 (0.00) | 1.00 (0.00) | 1.00 (0.00) |
| #pos training | 372.1 (33.83) | 348.6 (73.22) | 322.4 (83.10) | 322.0 (80.84) | 276.2 (119.34) |
| #neg training | 375.0 (0.00) | 375.0 (0.10) | 375.0 (0.17) | 375.0 (0.10) | 375.0 (0.10) |
| #pos test | 124.7 (11.32) | 116.7 (24.50) | 108.0 (27.81) | 107.8 (27.50) | 92.5 (39.90) |
| #neg test | 125.0 (0.00) | 125.0 (0.10) | 125.0 (0.17) | 125.0 (0.10) | 125.0 (0.10) |
| Count | 2087158074 | 11682341 | 1964898 | 413014 | 47582 |
| $c/v$ | 3.50 | 4.00 | 4.20 | 4.31 | 4.50 |
| $p/n$ training | 0.992 | 0.930 | 0.860 | 0.859 | 0.737 |
| $p/n$ test | 0.997 | 0.934 | 0.864 | 0.863 | 0.740 |

This section illustrates the performance of symbolic and deep neural network approaches. We generated sets of synthetic 3-CNF SAT formulas with 100 variables and 350, 400, 420, 431, and 450 clauses, respectively. Assignments of formulas were randomly divided into training (80%) and testing (20%) and used as an input dataset. For evaluation, we use the *accuracy* and *F1-Score* metrics and report positives, negatives, and number of solutions using Ganak (Sharma et al., 2019). Formulas with fewer clauses present an exponentially larger number of solutions than those with more clauses (Table 3). In this scenario, the search (learning) space is extensive, and the number of assignments is approximately equivalent for positives and negatives. The symbolic method using Valiant's Algorithm (VA) (Valiant, 1984) learning

procedure shows difficulties in generalizing from training to the test set (memorization of formulas). The *accuracy* of VA increases as the number of clauses increases; the mean value ranges from 62% (350 clauses) to 95% (450 clauses). This result is interesting because VA always reaches 100% training accuracy, but instead of learning, it memorizes the training data and generalizes poorly. Additionally, the ratio of positives and negatives solutions could direct this result because VA learns based on positives solutions in the learning space. Conversely, the neural network approach generalizes their learning better, independently of the number of clauses and the ratio of positives and negatives assignments.

## 4. Conclusions

The recent success of deep learning techniques over various domains and the positive theoretical results on the learnability of boolean formulas prompts the question of how effective DL approaches are in learning boolean formulas, which encode relevant problems in a variety of relevant domains, including symbolic reasoning and combinatorial optimization. We have tackled this subject by assessing the learning capabilities of multi-layer perceptrons (MLPs) – the basic components of deep learning systems – over boolean formulas encoding large model sampling benchmarks, combinatorial optimization problems, and random 3-CNFs with various constrainedness (clause-to-variable ratios). Our methodology consists of generating positive (satisfying) and negative (falsifying) examples for each formula, and verifying the cross-validation (CV) accuracy of a MLP as a means to assess its capability to learn the formula.

MLPs were very good approximators of all studied formulas, with no cross-validation accuracy below 99%. We thus use the more challenging measure of how many formulas were perfectly learned, i.e. with 100% CV accuracy, as a proxy for learnability. Our extensive experiments have four main findings:(i) neural learning generalizes better than pure rule-based systems (Appendix B) and pure symbolic approach (Sect. 3.3); (ii) relatively small and shallow neural networks can learn several families of boolean functions that encode combinatorial optimization problems (Sect. 3.1); (iii) smaller formulas seem harder to learn, possibly due to the fewer positive (satisfying) examples available; and (iv) interestingly, underconstrained 3-CNF formulas are more challenging to learn than overconstrained ones (Sect. 3.2). Our results can foster more empirical or new theoretical studies on the learning capabilities of deep learning models. For example, it would be interesting to further investigate whether the harder learnability of small formulas is associated with the better "coverage" of their datasets (since their size is fixed in our experiments, they are more sparse for large formulas), or some property of their factor graphs (as in Yolcu and Póczos (2019), where edges connect variables to clauses). Another line of future work could focus on gaining insights of what the neural networks have learned. This can be done via explainability methods (e.g. SHAP Lundberg and Lee, 2017) or the deduction procedure mentioned by (Valiant, 1984), where we would extract the formula learned by the MLP (using knowledge extraction methods, Tran and Garcez (2018)) and check to which extent it matches the original one.

**Acknowledgments**: Some experiments used the PCAD lab `http://gppd-hpc.inf.ufrgs.br` at INF/UFRGS. Work partly supported by Coordenação de Aperfeiçoamento de Pessoal de Nível Superior CAPES - Finance Code 001 and the Brazilian Research Council CNPq.

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

## Appendix A. Methodological details

We are interested in the learning capabilities of deep neural networks on classes of boolean formulas. For our purposes, a class is a set $S$ of formulas with certain characteristics. For example, a set with random 3-CNFs with 20 variables and 90 clauses, and another with 3-coloring problems on graphs with 30 vertices and 60 edges are distinct classes. Given a set $S$ of boolean formulas, we assess the learnability on each formula $f \in S$ by (i) creating a dataset containing positive (satisfying $f$) and negative (falsifying $f$) samples; and (ii) evaluating the performance of a deep neural network on this dataset. The two steps are detailed next.

**Dataset generation**  Our generation of negative samples from a formula $f$ is trivial: each variable is assigned either truth-value with 50% probability and we add the resulting assignment to the dataset if it falsifies $f$. Our boolean formulas of interest have much more negative than positive examples, hence this procedure is expected to take linear time on the desired number of negative samples and the generated samples are expected to be evenly distributed over the negative sample space. The generation of positive samples (models) for $f$ with the same properties requires the use of advanced model sampling techniques (Meel, 2014). The trivial procedure of calling a SAT solver sequentially to enumerate models yields samples poorly distributed over the model space: when a model with $t$ free variables is found, this procedure will generate $2^t$ models by assigning values to the free variables, where the remaining ones are unchanged.

Unigen2 (Chakraborty et al., 2015)[3] provably generates positive samples quasi-uniformly distributed over the model space. Moreover, Unigen2 handles large boolean formulas by leveraging their minimal independent support if it is known[4]. On formulas with less than 40 models, Unigen2 advises using the "trivial" procedure and enumerating all models, which we do with Glucose3 (Audemard and Simon, 2018). We aim to generate the same number of positive and negative samples, which vary according to our experiments (Section 3 on the main paper). The retrieved positive and negative samples of formula $f$ are then shuffled to generate its dataset. Dataset generation was done on a computer with 32-core (64 threads) 4xIntel Xeon X7550 CPU @ 2.00 GHz and 128 GiB DDR3 RAM.

---

3. We did not use the newer Unigen3 (Soos et al., 2020) as it was released during the execution of our experiments.

4. An independent support is a set containing variables that are sufficient to determine the value of the remaining ones (dependent support) on satisfying assignments. We changed Unigen2 so that it returns values for variables in the dependent support as well.

**Performance evaluation**   Theoretical studies on learnability are usually concerned with the hardness, in terms of computational complexity classes, of learning certain families of boolean formulas or worst-case sample complexity to achieve certain tolerance and confidence thresholds within the PAC learning framework. The focus of this study is conjunctive normal formulas with bounded number of literals per clause (k-CNFs).

Valiant (1984) demonstrated that k-CNFs are learnable in polynomial time, without mention of neural networks in particular. Hence our interest in showing empirical evidence of neural network learning capabilities on such formulas. To do this, given a CNF $f$, we generate a dataset with the aforementioned procedure and, without loss of generality, assess the learning capabilities of a deep neural network on $f$ via the $k$-fold cross-validation accuracy in the resulting dataset. This gives an individual (per-formula) measure of performance. To evaluate the learnability over a set $S$ of formulas belonging to a specific class, we use the average and minimum accuracy over the formulas in $S$. We also use the ratio of perfectly-learned formulas in $S$. That is, we count in how many formulas the $k$-fold cross-validation accuracy is 100% and divide by $|S|$. This is a stricter measure of performance that we use in our evaluations.

## Appendix B. Model sampling benchmarks

This section compares the performance of deep neural networks and decision trees on the model sampling benchmark of Unigen2 (Chakraborty et al., 2015). The benchmark formulas are usually large, containing up to hundreds of thousands of variables. They encode model checking problems, variations of SMTLib (Barrett et al., 2010) instances and problems arising from automated program synthesis. For each formula, a dataset with 500 positive and 500 negative instances was generated.

In this test, our deep NN is a Multi-Layer Perceptron (MLP) with two hidden layers containing 200 and 100 neurons, respectively. We pit MLPs against Decision Trees (DTs) on the premise that the DTs can competently learn boolean formulas as the resulting models are akin to Binary Decision Diagrams. DTs were executed with default scikit-learn parameters (Pedregosa et al., 2011), for comparison. Table 4 shows the average accuracy of a 5-fold CV for each formula.

The MLP perfectly learned all formulas generalising well from training to test folds, whereas the decision tree had a high accuracy overall, but did not fully learn any formula. DTs are known for overfitting data and this might be preventing them from generalizing from training to test folds. Moreover, the superior capabilities of MLPs is in line with recent findings (Ciravegna et al., 2021), where the authors have shown that, because of their better performance, it is better to add explainability capabilities to black-box neural networks than to directly use white-box models, of which DTs are an example. On the following experiments, we further evaluate MLPs on other classes of boolean formulas.

Table 4: Results (5-fold CV accuracy %) of MLP and DT on large boolean formulas from Unigen2 model sampling benchmark (Chakraborty et al., 2015). The MLP learns all formulas, and the DT approximates well but does not learn to perfection.

| Instance | #vars | #clauses | MLP | DT |
|---|---|---|---|---|
| s1238a_3_2 | 686 | 1850 | 100 | 97.12 |
| s1196a_3_2 | 690 | 1850 | 100 | 98.11 |
| s832a_15_7 | 693 | 2017 | 100 | 98.21 |
| case_1_b12_2 | 827 | 2725 | 100 | 98.21 |
| squaring16 | 1627 | 5835 | 100 | 98.01 |
| squaring7 | 1628 | 5837 | 100 | 98.41 |
| LoginService2 | 11511 | 41411 | 100 | 98.01 |
| sort.sk_8_52 | 12125 | 49611 | 100 | 97.52 |
| 20 | 15475 | 60994 | 100 | 97.61 |
| enqueue | 16466 | 58515 | 100 | 97.51 |
| karatsuba | 19594 | 82417 | 100 | 98.31 |
| llreverse | 63797 | 257657 | 100 | 97.13 |
| tutorial3.sk_4_31 | 486193 | 2598178 | 100 | 92.56 |

