# OpenReview forum: "Understanding Boolean Function Learnability on Deep Neural Networks: PAC Learning Meets Neurosymbolic Models"
_nesyconf.org/NeSy/2025/Conference_Phase_2 — NeSy 2025 - Phase 2 Oral_

### Official Review · Reviewer_Lz9M · 2025-07-04

**Rating:** 7
**Confidence:** 4

**Review:**

This paper explores the capability of deep neural networks to learn and generalize symbolic Boolean functions, focusing particularly on formulas expressed in conjunctive normal form (CNF). The authors design synthetic benchmarks, apply modern architectures (notably transformers), and examine the “learnability” of various Boolean concepts under neural models.

Pros
+ The paper tackles a fundamental and underexplored question: Can deep learning (DL) models capture symbolic reasoning, particularly logic-based concepts like CNFs? This direction is valuable for bridging neural and symbolic reasoning.  The synthetic datasets and learnability experiments are thoughtfully constructed to test specific hypotheses.

+ The experimental results suggest that DL models can indeed learn Boolean formula behaviors, with transformers outperforming baselines. This work contributes to discussions about integrating logic and neural models, and may inspire further studies in neurosymbolic AI.

Cons
- Presentation Issues: The acronym CNF is not expanded on first use (until page 3), which can be confusing to readers not deeply familiar with logic formalisms. Section title “Learnability experiments” should be capitalized to match the formatting of other section titles.

- While results show promising performance, the paper would benefit from a detailed failure case analysis. For example, are there specific Boolean structures or lengths for which models fail? Are there consistent patterns in the errors?

- The findings demonstrate what works but fall short in explaining why—e.g., whether the models are truly learning symbolic patterns or just statistical correlations.

**Anonymity:**

Remain anonymous

---

### Official Review · Reviewer_YQfZ · 2025-07-08
**Interesting but with room for improvement**

**Rating:** 6
**Confidence:** 3

**Review:**

The paper addresses the problem of how Boolean formulas can be learned by deep neural networks in practice. In their work, the authors investigated the learning capabilities of multi-layer perceptrons (MLPs) on Boolean formulas encoding model-sampling benchmarks, combinatorial optimization problems, and random 3-CNFs with varying degrees of constrainedness.

The experimental findings are interesting and provide a further contribution to the development of neurosymbolic methods. However, the structure of the paper could be improved, as well as the explanation of some of its parts.

Here are some comments and suggestions:
1. Provide first and last names of Bengio, Hinton, and LeCun in Sec. 1.
2. "Artificial Neural Networks (ANNs)" capitalized in Sec. "Related Work" (also capitalized, as for the other sections).
3. Correct the form "symbolic and neural approaches' learning" as well as "scikit-learn's implementation" in Sec. 3.
4. Correct the typo "We have have..." in Sec. 3.
5. No need to reintroduce acronyms such as MLPs and DTs several times.
6. Briefly explain what $\beta_1$ and $\beta_2$ denote in Sec. 3.
7. Fix the sentence about "3-CNF PAC-learnable algorithm" in Sec. 3, because there seems to be a missing part.
8. The first sentence of Sec 3.1 refers to something that is not in the previous section.
9. Briefly explain what flat and morphed graphs are.
10. Briefly explain the Brelaz heuristic.
11. Add a whitespace between "%perfect" on Page 6.
12. Correct the typo "overconstrained formulas" on Page 9.
13. Briefly explain the Ganak probabilistic exact model counter.
14. Briefly explain the Valiant's Algorithm (VA).

**Anonymity:**

Remain anonymous

---

### Official Review · Reviewer_7mNU · 2025-07-08
**Review: Understanding Boolean Function Learnability on Deep Neural Networks**

**Rating:** 7
**Confidence:** 3

**Review:**

The paper addresses Boolean learnability and explores multiple dimensions of the problem.
The experimental analysis in Section 3.2 focuses on the learning performance when varying (i) the size of the Boolean formulae (e.g., up to 100 variables), and also (ii) the clause-to-variable ratio. Further, the authors address an interesting question: how satisfiability hardness relates to learnability in random 3-CNF formulas.
The finding that underconstrained formulas are harder to learn than overconstrained ones is a novel (and perhaps counter-intuitive).

Clarity: Well-written paper.

Significance: The paper shows to which extent Boolean learnability can happen in practice in standard neural network models. While this might be an understudied topic (as the authors mention), the results of this paper can offer new insights and even challenge assumptions that might be carried over from the domain of SAT solving into the broader NeSy field.

**Anonymity:**

Remain anonymous